# Relationship between the Young’s Moduli of Whole Microcapsules and Their Shell Material Established by Micromanipulation Measurements Based on Diametric Compression between Two Parallel Surfaces and Numerical Modelling

**DOI:** 10.3390/mi14010123

**Published:** 2023-01-01

**Authors:** Daniele Baiocco, Zhihua Zhang, Yanping He, Zhibing Zhang

**Affiliations:** 1School of Chemical Engineering, University of Birmingham, Edgbaston, Birmingham B15 2TT, UK; 2Changzhou Institute of Advanced Manufacturing Technology, Changzhou 213164, China; 3School of Chemical Engineering, Kunming University of Science and Technology, Chenggong Campus, Kunming 650504, China

**Keywords:** plant-based, non-synthetic, microcapsules, micromanipulation, intrinsic mechanical properties, apparent elastic modulus, mathematical modelling, finite element analysis

## Abstract

Micromanipulation is a powerful technique to measure the mechanical properties of microparticles including microcapsules. For microparticles with a homogenous structure, their apparent Young’s modulus can be determined from the force versus displacement data fitted by the classical Hertz model. Microcapsules can consist of a liquid core surrounded by a solid shell. Two Young’s modulus values can be defined, i.e., the one is that determined using the Hertz model and another is the intrinsic Young’s modulus of the shell material, which can be calculated from finite element analysis (FEA). In this study, the two Young’s modulus values of microplastic-free plant-based microcapsules with a core of perfume oil (hexyl salicylate) were calculated using the aforementioned approaches. The apparent Young’s modulus value of the whole microcapsules determined by the classical Hertz model was found to be *E_A_* = 0.095 ± 0.014 GPa by treating each individual microcapsule as a homogeneous solid spherical particle. The previously obtained simulation results from FEA were utilised to fit the micromanipulation data of individual core–shell microcapsules, enabling to determine their unique shell thickness to radius ratio (*h/r*)*_FEA_* = 0.132 ± 0.009 and the intrinsic Young’s modulus of their shell (*E_FEA_* = 1.02 ± 0.13 GPa). Moreover, a novel theoretical relationship between the two Young’s modulus values has been derived. It is found that the ratio of the two Young’s module values (*E_A_*/*E_FEA_*) is only a function on the ratio of the shell thickness to radius (*h/r*) of the individual microcapsule, which can be fitted by a third-degree polynomial function of *h/r*. Such relationship has proven applicable to a broad spectrum of microcapsules (i.e., non-synthetic, synthetic, and double coated shells) regardless of their shell chemistry.

## 1. Introduction

Nowadays, microcapsules are being incorporated in many industrial fast-moving consumer goods (FMCG), with particular emphasis on laundry formulations and functionalised textiles [1]. Interestingly, microcapsules have proven effective at minimising the amount and then the cost of fragrance actives required in formulation [2], since they act as protective carriers of the active [3], thereby boosting its physico-chemical stability and shelf-life [4]. Perfume microcapsules (PMCs) for laundry applications are aimed to selectively deliver the fragrance active onto fabrics following their deposition during a washing-drying cycle [5,6]. When a mechanical action is applied by rubbing or caressing dry garments, PMCs can release their perfuming load for an enhanced costumer experience [3]. Over the last decades, melamine formaldehyde (MF) has been the leading shell material for the fabrication of PMCs [7], due to its desirable properties (e.g., water resistance and great acid-base tolerance) and relative inexpensiveness at an industrial scale [3]. Although a broad variety of PMCs can be efficiently prepared with MF [8], outstanding environmental, health & safety (EHS) concerns have arisen around microplastic-based (melamine) and presumably health-threatening (formaldehyde) materials [9]. Increased self-awareness against the potential carcinogenic effects and poor indoor air quality due to formaldehyde has been raised [10]. Moreover, adverse environmental impact of released formaldehyde has been reported, although its residue in products is still within the legal limit [11]. As a first step towards complying with the regulations enforced against carcinogenic materials, novel coating materials have been attempted for laundry and textile applications, such as polysulfone [4], polyurethanes-urea and polyesters [12]. Although the above mentioned materials can provide several desirable features (e.g., thermal stability and mechanical resistance) [13], they are costly (raw materials and processing) [14], poorly/non-biodegradable, and contain respiratory toxic and asthma-inducing isocyanates [15], which highly restrict their potential applications. Therefore, intense efforts towards encapsulating fragrance oils using natural polymers have been undertaken. Bruyninckx and Dusselier [16] have reviewed several sustainable and high performance alternatives for encapsulation of volatile organic compounds (VOCs). Hazard-free gelatine (Gl), chitosan (Ch), and gum Arabic (GA) can form the shell around VOCs by complex coacervation yielding high payloads [17,18]. However, the animal origin of Gl and Ch may still affect their global acceptance in consumer products following the pathogenesis of novel diseases, such as human prion protein misfolding [19]. As a further step towards overcoming MF-related concerns and the biodegradability issues of synthetic polymers, as well as complying with personal and religious beliefs, microplastic-free plant-based microcapsules from safe biopolymers are currently being developed [2]. Therefore, in-depth understanding of the intrinsic mechanical properties of the shell materials of such microcapsules including the Young’s modulus is crucial to assess their performance during fabrication and at potential end-use applications [20]. 

As reported in the literature, the application of a load onto individual microcapsules enables to generate compression force versus displacement profiles [21]. Interestingly, these can be further analysed to determine the mechanical properties, including the Young’s modulus, which is essential to predict the deformational behaviour of a material when facing an external force. 

Several system-specific techniques for the mechanical characterisation of objects have arisen over the years, including strip-ring extensiometry for hydrogels [22], micropipette aspiration for cellular elasticity [23], microinjection of biological cells [24], optical tweezers for human red blood cells and lipid membranes [25,26], microelectromechanical systems (MEMS) for investigating the biophysical properties of human breast cancer cells [27], and microfluidic channels for liquid-loaded capsules [28]. Notwithstanding, atomic force microscopy (AFM) and micromanipulation via compression between two parallel surfaces are still the preferred techniques owing to their adaptability, and broad range of applicable loads [20,21]. AFM has proven suitable for investigating the mechanical behaviour and Young’s modulus of many biological cells, and active-laden microcapsules. Pinpoint forces can be exerted with superb precision by AFM, which detects deformations in the range of a few nanometres [29]. Specifically, the deformational behaviour of soft particles in food science [30], microbial cells with a core–shell configuration [31], and biofilm build-ups [32] have been investigated by AFM, as well as the biomechanical properties of cellular membranes during mitosis and ligand-interacting membrane proteins [33], and the nanomechanical response of trapped bacteria or viruses [34]. The apparent Young’s modulus of microcapsules (*E_A_*) entrapping phase change materials (PCMs) within acrylate shells was also quantified by AFM. This was fulfilled by employing a SiO_2_ probe with a Young’s modulus value E_SiO2_~75 GPa (baseline). Considering the baseline and the response generated by the pinpoint probe onto a specific area of the microcapsule shell, the apparent Young’s modulus of microcapsules was evaluated by the Hertz model, which yielded values with a significant variability (~0.15–1.5 GPa) [35]. However, the Young’s modulus values of microcapsules determined using the AFM measurement combined with the Hertz model should be interpreted with caution since they can depend on the penetration depth of the tip attached to the cantilever (nanoindentation). The greater the penetration depth, the more impact the liquid core can generate on the measurement results. Moreover, since AFM tends to measure local mechanical behaviours of materials, the results may be greatly affected by inhomogeneous and non-smooth shells, especially when the shell thickness and the liquid reservoir depth of a microcapsule are unknown. Another general disadvantage of AFM is mainly due to its incapability in covering more extended surface areas at a time, regardless of the tip geometry (e.g., spherical and conical shape) [21]. In the literature, there are various reports on the Young’s modulus of microcapsules. For example, the instantaneous Young’s modulus from Poly(d,l-lactide-co-glycolide) (PLGA) based microcapsules was measured by Sarrazin et al. [36] at minute indentation depths (0–12 nm) with significant viscoplastic effects being observed from an indentation of ~5 nm. The Young’s modulus values of poly(styrene sulfonate) –poly(allylamine) microcapsules and poly(urea–formaldehyde) composite microcapsules [37] were found to be 2.5–4.0 GPa. Moreover, it is difficult to maintain the alignment between the AFM colloid probe and spherical microcapsules which can then slip away easily [20]. 

In contrast, micromanipulation has proven more effective for the compression of single microparticles including microcapsules with a core–shell configuration between two parallel surfaces, since a typical force range (100 nN–1.0 N) greater than that of AFM (from pN to μN) can be applied [21,38]. Moreover, it allows displacements greater than the object of study (e.g., size of microcapsule) to be generated [39]. Furthermore, micromanipulation can yield the rupture of individual particles under compression, which is difficult to achieve using other techniques [40].

Interestingly, the resulting force-displacement data can be fitted to specific models to estimate both apparent and intrinsic Young’s modulus of particles, which is conditional upon their structural configurations (e.g., solid microbeads and core–shell microcapsule) [5,41]. Accordingly, the apparent Young’s modulus of relatively porous calcium-shellac microbeads was estimated by the Hertz model (0.54 ± 0.09 GPa) [42,43], whereas the intrinsic Young’s modulus poly(methyl-methacrylate) microcapsules with a core–shell configuration was quantified via a finite element model (0.75 ± 0.3 GPa) [6]. Over the years, simplistic mathematical solutions have been proposed to determine the Young’s modulus of microparticles/microcapsules. Smith et al. [44] first pioneered the application of a model based on finite element analysis (FEA) to compression experiments in order to determine the mechanical properties of fully elastic cellular walls at different fractional deformations (*ε*). However, when dealing with core–shell microcapsules, a fully elastic behaviour of their shell cannot be assumed, especially at high fractional deformations [45]. Core-shell MF microcapsules have proven plastic deformation at *ε* ≥ 0.2, which may not rupture until *ε* ≥ 0.5 [46]. Indeed, any stretching, bending, and wrinkling effect of the shell under compression should be taken into account. Interestingly, Mercade-Prieto et al. [3] have developed a powerful FEA model for core–shell microcapsules, which is capable of evaluating the intrinsic Young’s modulus of the shell materials (*E_FEA_*), as well as calculating the unique shell thickness to radius ratio (*h/r*) of individual elastoplastic microcapsules using the force versus displacement data corresponding to relatively low fractional deformation (*ε* ≤ 0.1 within the elastic region). This model relying on ad-hoc *h/r*-dependent polynomial functions (Table 1) has proven highly accurate with its output *h/r* values being validated against a large number (186) of cross-sectional images of individual microcapsules via Transmission Electron Microscopy (TEM) [5]. The establishment of a specific core–shell model has represented a breakthrough over the classical interpretation of the apparent Hertzian Young’s modulus (*E_A_*) which can approximately address microcapsules with a liquid core as whole solid-like particles. With that being said, for the same microcapsules, two different Young’s modulus values can be obtained using the Hertz model and FEA based on the same force versus displacement data obtained by micromanipulation, which can differ by an order of magnitude [47]. To the authors’ best knowledge, their interrelationship has not yet been established. The present study therefore aims to develop and establish a theoretical relationship between the apparent Young’s modulus (*E_A_*) of single whole microcapsules calculated by using the Hertz model, and their intrinsic Young’s modulus (*E_FEA_*) of the shell material determined using FEA, which is intimately related to the *h/r* value of microcapsules. The experimental force versus displacement data were generated using an advanced micromanipulation technique based on diametrical compression of individual microcapsules between two parallel surfaces. Ad hoc simulations were run to validate the results. As presented in our previous papers [2,48], microplastic-free biopolymer-based microcapsules with a core of perfume oil (hexyl salicylate) have been employed herein, and their micromanipulation dataset was used for direct validation. In addition, different microcapsules with a synthetic and composite shell were also used to investigate the broad spectrum applicability of the novel model regardless of their microencapsulation process and/or shell chemistry. This results can be used to interpret various Young’s modulus data of microcapsules reported in the literature and the approach taken represents a unique and unambiguous methodology to characterise the elastic properties of microcapsules with a core–shell structure.

## 2. Materials and Methods

### 2.1. Materials

Gum Arabic and fungal chitosan (fCh; deacetylation degree 79%, molecular weight ~150 kDa) were purchased from Nexira Food (Rouen Cedex, France, EU) and Kitozyme (Herstal, Belgium, EU), respectively. Analytical grade chemicals being hexylsalicylate (HS; 1.04 g·mL^−1^), sorbitan triolate (Span85), triethanolamine (TEA), aqueous glutaraldehyde (GLT; 50% *w*/*w*), fuming hydrochloric acid (HCl; 36% *w*/*v*), LR white acrylic resin were bought from Sigma-Aldrich (Gillingham, Dorset, UK), stored according to the safety data sheet (SDS) instructions, and used without any additional purification. All admixtures were prepared with demineralised water (18.2 MΩ·cm at 25 °C).

### 2.2. Preparation of Microcapsules

The microcapsules were fabricated via one-step complex coacervation according to Baiocco et al. [2]. Briefly, HS (40 g) dyed with fluorescence sensing Nile Red (~5 mg) was added to an aqueous admixture (730 mL) at pH 1.95 (acidified by HCl_aq_) containing GA (2.0% *w*/*v*) and fCh (0.5% *w*/*v*), which led to two phases. Sorbitan triolate (0.8 g) as the emulsifier was added. Homogenisation (1000 rpm; IKA Eurostar 20, Germany, Staufen, EU) was carried out to achieve oil-in-water (o/w) droplets with a target mean size of ~30 µm measured by laser diffraction. Complex coacervation between GA and fCh was induced by increasing the pH to 3.4 via gradual addition (160 mL) of TEA until a shell encircling the oil droplet was evident under a bright-field microscope (Leica DM500, Buffalo Grove, IL, USA). GLT (0.3 g/g-biopolymer) was added to trigger the crosslinking with amines along the microcapsule shells, hence their reticulation. The suspension of microcapsules was left crosslinking under stirring (300 rpm; ~15 h). 

### 2.3. Mechanical Properties

A micromanipulation technique based on parallel compression of individual microcapsules was utilised to determine the mechanical properties of single microcapsules [21,42,46]. A flat glass slide (~2.5 cm^2^; thickness ~ 1.5 mm) was covered with two drops of suspended microcapsules, which were then left to dry out at ambient temperature (23.5 ± 1.5 °C). Microcapsules were observed by a side-view camera (high performance charge-coupled device camera, Model 4912-5010/000, Cohu, Poway, CA, USA). A flat-end glass probe was attached to a force transducer (Model 403A, force scale 5 mN; Aurora Scientific Inc., Aurora, ON, Canada) with a measurement resolution of ±0.1 μN. The probe was enabled to move down/upwards by a fine micromanipulator with a displacement resolution (± 0.1 µm). A descending speed of 2.0 μm·s^−1^ was selected to compress each particle. Thirty randomly chosen microcapsules were compressed in order to generate statistically representative results [46]. Since particles can exhibit elastic, viscoelastic, and elastoplastic shells, their behaviour should be determined through compression-holding-unloading experiments and different speeds. Previous literature has demonstrated that polymeric excipients [49] and thin-shell MF-based microcapsules [50] with different cores exhibited mainly elastoplastic deformations [46]. Moreover, a negligible viscous character was reported for GA, thereby it mainly exhibited elastic deformations [51]. Similarly, Ch has proven elastic properties [52], especially when combined with xanthan gum [53]. Accordingly, only one compression velocity (2.0 μm·s^−1^) was used to compress the microcapsules. 

### 2.4. Morphology

Bright-field optical (PL-Fluotar 5×/0.12 and 10×/0.30 lens) and fluorescence (Cool-LED pE-300 blue light beam, wavelength λ = 460 nm) microscopy, as well as accelerated voltage (15–30 kV) scanning electron microscopy (SEM; JEOL 6060, Peabody, MA, USA) were employed to assay the microcapsules for their morphology, surface topography, and structural properties. 

### 2.5. Particle Sizing

Laser diffraction technique was used to quantify the mean moist diameter (i.e., volume-to-surface Sauter diameter D_[3,2]_) and size distribution of the microcapsules (Mastersizer2000, Malvern Instruments, Malvern, UK). The microcapsule suspension (~3 g) was added into the sample dispersion unit under stirring (2500 rpm) coupled with the instrument. The tests were performed at ambient temperature employing a He–Ne laser (measurement range of 50 nm–0.9 mm). The number-based diameter (D_n-b_) of dry microcapsules was measured via on-screen image analysis following calibration of the micromanipulation side view camera with a calibrating eyepiece graticule slide (10 µm microcalibrating ruler, Graticules Ltd., Tonbridge, Kent, UK) [40].

## 3. Results and Discussion

### 3.1. Morphology

Figure 1A displays HS-laden microcapsules fabricated by CC. Most of microcapsules encircled single oil droplets (yolk-white like structures) within their fungal chitosan-gum Arabic (fCh-GA) shells. The microcapsules appeared to be individual, hence neither clustering nor agglutination phenomena among the microcapsules were observed. Interestingly, a morphological analysis revealed the presence of relatively spherical microcapsules. Specifically, slightly elongated shells with an eye-shape were observed, which has been similarly reported by Leclercq et al. [17] for gelatine based microcapsules encapsulating limonene. This eye-shaped configuration could be ascribable to fast stirring while inducing coacervation, thereby triggering an alignment of the forming shells (mobile shells) of microcapsules with the flow pattern within the agitated vessel. Alternatively, Baiocco et al. [2] have elucidated that some excess polymeric material could be deformed around the oil droplets by the agitation during the development of the shell, hence generating the eye shape. Surface topography was investigated by SEM observation, which highlighted the presence of HS-microcapsules with a relatively smooth shell [48]. However, several rough and indented areas were also visible at the shell surface (Figure 1B,C). Specifically, multiple surface vacuoles were observed. This might be a result of the high vacuum inside the SEM chamber, which affected the structural chassis of microcapsules, according to Farshchi et al. [54]. Thus, such surface vacuoles might act as inside-out bridges for the core oil (HS) to suddenly diffuse out through the shell in the chamber of the scanning electron microscope (Figure 1C). 

### 3.2. Particle Sizing

The microcapsules were assayed for their Sauter diameter and size distribution (Appendix A). They were determined to range between 14 µm and 100 µm (D_[3,2]_ = 36.5 ± 1.2 µm) with a SPAN value of 1.1, which is consistent with our previous works [2,48,55]. Statistical analysis suggested that the size distribution could be fitted to a lognormal distribution curve with 95% confidence. In addition, the size of microcapsules detected by laser diffraction was found to be in good agreement with their SEM micrographs (Figure 1B,C).

Figure 2 presents the optical images of the wet microcapsules which were left air-drying (25 ± 1.5 °C) whilst being monitored via real-time optical microscopy imaging before (0 h; Figure 2(A1–C1)) and after full drying (~1 h; Figure 2(A2–C2)). In moist conditions the spherical microcapsule (A1) exhibited a moist diameter of 22.0 ± 0.7 µm whilst it was found to be 17.6 ± 0.5 µm (~20% smaller) under dry conditions. When dealing with B1 and C1, the size measurements were conducted along the cross-sectional diameters of each microcapsule. As can be seen, the shell material was formed abundantly around the oil droplets, which might therefore trap a great number of minute water molecules within [56]. The moist diameters of Figure 2(B1,C1) were determined to be 14.5 ± 0.3 µm and 15.3 ± 0.2 µm, respectively. As anticipated, their dry diameters were 20–25% lower, being 10.6 ± 0.3 µm and 12.1 ± 0.4 µm, respectively. Moreover, Figure 2(A3–C3) display the fluorescence sensing response of the three microcapsules in dry conditions. The photomicrographs clearly elucidate the effective retention of the perfume oil emitting a green visible spectrum when being excited by the fluorescent light source. This phenomenon is attributable to the solvatochromatic response of the dye (i.e., Nile Red) within a relatively polar environment (i.e., hexyl salicylate), as also discussed by Baiocco et al. [55] and Zhang et al. [43]. The corresponding diameters obtained from the fluorescent micrographs were 16.9 ± 0.7 µm, 10.2 ± 0.2 µm and 10.5 ± 0.3 µm for Figure 2(A3,B3,C3), respectively, which appeared not to be statistically different from those attained under optical light in dry conditions for the same microcapsules (Figure 2(A2,B2,C2), with 95% confidence. Clearly, it was not possible to estimate the shell thickness via direct comparison between the bright filed and fluorescence photomicrographs. However, this result seems to suggest that the perfume oil might possess some natural affinity for the coacervate network, possibly migrating into the shell via partial solubilisation, thereby enabling the fluorescent signal to be detected within the shell as well. Overall, it could be inferred that the shell material retained some water leading to a partial moisturisation/swelling of the shell (moist diameter), which shrank to a certain extent upon drying.

### 3.3. Mechanical Properties

#### 3.3.1. Apparent Young’s Modulus of Whole Microcapsules Determined by the Hertz Model

The classic Hertz model has proven effective at describing the relationship between the compression force (*F*) and the axial displacement (*δ*) of a spherical particle at minute deformations (≤10%) [38]. The model assumes (i) frictionless and smooth contact surfaces, (ii) homogeneous, isotropic and linearly elastic (Hooke’s law) contacting materials, and (iii) negligible geometric non-linearities due to larger strains [47]. While being aware of the aforementioned assumptions, the Hertz model has been applied to determine the apparent Young’s modulus (*E_A_*) of core–shell microcapsules including their liquid core, which are treated as solid spheres:(1)$$F=\frac{\psi \;}{1-\nu^{2}}E_{A}r^{2}{\left(\frac{\delta }{2r}\right)}^{3/2}$$
where *ψ* is the spherical shape factor equal to 4/3, *ν* is the Poisson ratio that is assumed to be 1/2 for non-compressible polymeric microcapsules, whereas the group *δ*/(2*r*) represents the fractional deformation (*ε*). An example of the Hertz model fitting to the force versus displacement data obtained from micromanipulation for a single microcapsule is illustrated in Figure 3A, whereas the mean apparent *E_A_* value along with the corresponding mean coefficient of determination (R^2^) of the model performance for the 30 tested microcapsules can be found in Table 2. The mean R^2^ value from all the microcapsules was determined to be ≥0.93, which indicates that the compression force versus displacement data of single microcapsules can be fitted by the Hertz model reliably [57]. Figure 4A-(i) displays the apparent (E_H_) Young’s modulus of microcapsules as a function of their diameter. Interestingly, the apparent Young’s modulus of the microcapsules did not seem to vary with the diameter significantly on average with 95% confidence (mean value *E_A_* = 0.095 ± 0.014 GPa). However, *E_A_* cannot fully represent the Young’s modulus of the pure shell material since the microcapsule contains a liquid core which in theory has no elasticity but can also contribute the force response under compression. In light of the above, it has become increasingly imperative to develop a methodology to overcome the shortcomings of the basic Hertzian theory, therefore predicting the intrinsic elastic modulus value of the shell material [5].

#### 3.3.2. Determination of the Shell Young’s Modulus of Microcapsules by FEA

In order to determine the intrinsic Young’s modulus value of the shell material of microcapsules, the FEA model developed by Mercadé-Prieto et al. [5] includes both bending and stretching of the microcapsule shell under compression. Mathematically, it is presented as follows:(2)$$F=E_{FEA}\frac{h}{r}\left[\;f_{1}\;{\left(\frac{\delta }{2}\right)}^{2}+f_{2}\;\left(\frac{\delta }{2}\right)r+f_{3}\;r^{2}\right]$$
where *E_FEA_* is the Young’s modulus of the shell material of microcapsules, *F* is the experimental compression force measured by micromanipulation, *δ* is the compressive axial displacement during compression, *h/r* is the ratio of shell thickness to the microcapsule initial radius, and *f*_1_, *f*_2_, and *f*_3_ are polynomial functions of *h/r* detailed in Table 1 [5].

A typical example of FEA model fitting to the force versus fractional deformation (*ε*) data of an individual microcapsule is presented in Figure 3B (R^2^ = 0.98). Figure 4A-ii displays the FEA-derived shell Young’s modulus of microcapsules versus their diameter, which did not seem to change with diameter significantly. When analysing the data, several non-aligned points were obvious at a given diameter. Specifically, at a diameter of 18 µm and 24 µm, the data scattered vertically between 0.9–1.5 GPa and 0.4–1.6 GPa, respectively. However, the most significant variability in the Young’s modulus values was found at a diameter similar to the mean size of microcapsules (~28 µm). Interestingly, the data ranged vertically from 0.15 GPa to 1.9 GPa, highlighting a variability of more than one order of magnitude. These findings likely suggest that the coacervate matter may form inhomogeneously around the oil droplets during microencapsulation, hence a direct effect on the intrinsic mechanical properties of the resulting shell material may be plausible (Figure 2A,B). When compared to the Young’s modulus values determined by the Hertz model, there appears to be an overall upward shift of the data points obtained via FEA, including their vertical scattering at each given diameter. Clearly there is a significant difference in the mean value of the two Young’s moduli determined by the two approaches. As mentioned, the Hertz model includes no *h/r* parameter, thus any physical difference between thin- and thick-shell microcapsule is difficult to predict. On average, the Young’s modulus by FEA from thirty HS-microcapsules was determined to be *E_FEA_* = 1.02 ± 0.13 GPa (with a corresponding unique (*h/r*)*_FEA_* = 0.132 ± 0.009), which is in line with that reported by Mercadé-Prieto et al. [5] for MF-based microcapsules via FEA (1.6 ± 0.3 GPa) [5]. Although these values appear to be statistically similar, the slight discrepancy is likely ascribable to the nature of the different shell materials. Specifically, the coacervate shell made of natural biopolymers (fChGA) may possibly form non-uniform and microporous structures, as also described by Espinosa-Andrews et al. [58]. In contrast, MF can form thin and highly smooth shells ((*h/r*)_MF_~0.02 ± 0.01 [5]), as with many other plastic materials [46]. Furthermore, this value (*E_FEA_* = 1.02 ± 0.13 GPa) is also in line with that of other polymeric microspheres for pharmaceutical applications (1.6 ± 0.2 GPa) investigated by Yap et al. [49]. In addition, similar values of the Young’s modulus were also obtained from other types of microcapsule shells utilising AFM, which is a different technique from micromanipulation. Specifically, core–shell microcapsules formulated with aminoplast [59], poly(styrene sulfonate)/poly(allylamine) [60], and poly(D,L-lactide-co-glycolide) [36] exhibited a Young’s modulus of to 1.7 GPa, 1.3 ± 0.15–1.9 ± 0.2 GPa, and 0.1–3.0 GPa, respectively. As discussed above, the Young’s modulus value from AFM, although it is also determined by the Hertz model, tends to represent the local stiffness of the test material near the surface (due to a typical indentation depth ~20–200 nm), the resulting Young’s modulus value obtained should be close to the intrinsic Young’s modulus of the shell. Notwithstanding, it may be difficult to predict any effect of the liquid reservoir (beneath the shell) on the AFM measurements when the shell is particularly thin, as with synthetic shells whose thickness can be as low as 20–70 nm [5]. Tan et al. [61] conducted AFM measurements on oil-laden microcapsules made of thiolated chitosan tentatively resulting in an apparent Young’s modulus of 1.44 MPa, which is surprisingly around three orders of magnitude lower than ours (1.02 ± 0.13 GPa). This major discrepancy was probably due to the combination of several effects, including the processing conditions (i.e., ultrasonic synthesis), and the chemistry of chitosan employed which had been grafted with thiol groups using DL-N-acetylhomocysteine thiolactone [61]. Moreover, a crucial role may also be played by the size of the spheres (<10 µm) and their extremely thin shells (<180 nm) leading to a *h/r*~0.018 which is one order of magnitude lower than our (*h/r*)*_FEA_* = 0.132 ± 0.009. Figure 4B displays the FEA-predicted *h/r* of microcapsules versus their diameter from compression experiments. It is found that *h/r* did not vary significantly with the diameter with 95% confidence.

#### 3.3.3. Interrelationship between the Apparent Young’s Modulus (EA) of Whole Microcapsules Determined Using the Hertz Model and That of the Shell Material Using FEA

As shown in Table 2, *E_FEA_* is clearly higher (approximately by one order of magnitude) than the corresponding *E_A_*. Such discrepancy is not surprising as the latter is based on treating a microcapsule as a homogenous microsphere and the Young’s modulus of a liquid core (or a fluid) is nil. By considering the two Young’s moduli can be determined using the same set of experimental data, it may be possible to establish their relationship mathematically. Given that the fractional deformation is *ε* = *δ*/(2*r)*, the model proposed by Mercadé-Prieto et al. [5] (Equation (2)) can also be expressed as a dimensionless force $\tilde{F}=F/E_{FEA}rh$:(3)$$\frac{F}{E_{FEA}rh}=f_{1}\varepsilon^{2}+f_{2}\varepsilon +f_{3},\;\;\;\;\;\;\;\left\{0.03<\varepsilon <0.1\right\}\;$$
where *f*_1_, *f*_2_ and *f*_3_ are the polynomial functions of (*h/r*), see Table 1 [5]. For a given *h/r* (e.g., *h/r* =0.05; 0 < *h/r* ≤ 0.14), Equation (3) can be used to generate the $\tilde{F}$ data corresponding to fractional deformations *ε* from 0.03 to 0.1 at progressively high steps (e.g., step-by-step threshold of 0.001). Five examples of the generated dimensionless force versus fractional deformation data (up to 10% deformation) for a given *h/r* are shown in Figure 5A. If the data is plotted in terms of the dimensional force $\tilde{F}$ versus *ε*^3/2^, see Figure 5B, their relationship looks approximately linear, and the slope increases with *h/r*, which is similar to Equation (1). Therefore, Equation (3) can also be expressed as follows:(4)$$\frac{F}{E_{FEA}rh}=f_{4}\varepsilon^{3/2}\;$$
where *f*_4_ is a function of *h/r* only. For a given *h/r* of 0.05, the value of *f*_4_ can be determined by linearly fitting the corresponding curve using Equation (4), which gives 0.549 with a coefficient of determination R^2^ = 0.997 (See the Appendix A). By comparing Equation (1) with Equation (4), the ratio of *E_A_*/*E_FEA_* defined by *φ* is given by:(5)$$\varphi =f_{4}\;\frac{1-\nu^{2}}{\psi }\;\frac{h}{r}\;$$

The corresponding *φ* value can therefore be determined to be 0.0154 for this particular value of *h/r*, which includes the Poisson ratio (*ν*) and the spherical shape factor (*ψ*).

Accordingly, increasingly high *h/r* values ((*h/r*)_k_ = 0.01, 0.02, …, 0.14) were chosen to generate the series of the corresponding *f*_4,k_ values and therefore *φ*_k_ (Figure 6).

Interestingly, *f*_4_ increases with *h/r* monotonically, and the resulting relationship is therefore fitted by a second-degree polynomial:(6)$$f_{4}=k_{1}{\left(\frac{h}{r}\right)}^{2}+k_{2}\left(\frac{h}{r}\right)+k_{3}\;$$
where *k*_1_ = 8.4673, *k*_2_ = 2.5728, and *k*_3_ = 0.1597 are the dimensionless constants of the polynomial, with their mean coefficient of determination of 0.999. Thus, the dimensionless force can be expressed as:(7)$$\tilde{F}=\left[k_{1}{\left(\frac{h}{r}\right)}^{2}+k_{2}\left(\frac{h}{r}\right)+k_{3}\;\right]\varepsilon^{3/2}\;$$
which leads to the following explicit equation between force (*F)* and displacement (*δ*) by substituting *ε*^3/2^ = (*δ*/2*r*)^3/2^ and $\tilde{F}=F/\left(E_{FEA}rh\right)$:(8)$$F=E_{FEA}\left[k_{1}{\left(\frac{h}{r}\right)}^{3}+k_{2}{\left(\frac{h}{r}\right)}^{2}+k_{3}\left(\frac{h}{r}\right)\;\right]{\left(\frac{\delta }{2}\right)}^{3/2}r^{1/2}$$

Since the compression force *F* is equal in both models (i.e., the Hertz and FEA) for a given fractional deformation, the combination of Equations (1), (5) and (8) thus leads to:(9)$$\frac{E_{A}}{E_{FEA}}=k_{1}{\left(\frac{h}{r}\right)}^{3}+k_{2}{\left(\frac{h}{r}\right)}^{2}+k_{3}\left(\frac{h}{r}\right)\;.\;$$
which is an explicitly third-degree polynomial function of *h/r* with 0 < *h/r* ≤ 0.14. Applying Equation (9), *E_A_*/*E_FEA_* was determined to be 0.085 ± 2 × 10^−3^ for HS laden microcapsules with a fCh-GA shell. For given values of *E_A_* and *h/r*, the predicted intrinsic *E*_*FEA**_ was 1.09 ± 0.14 GPa, which have no significant difference from those calculated directly via FEA (*E_FEA_* = 1.02 ± 0.13 GPa by Equation (1)) with 95% confidence. These results confirm the effectiveness of the model herein developed. Nonetheless, a general validation of the model is required in order to verify its prediction capability independently of the chemistry of the microcapsules. Having said that, the newly established Equation (9) was also applied to the micromanipulation data obtained from core–shell melamine-gluteraldehyde-formaldehyde (MGF) microcapsules with a core of perfume oil (i.e., lily oil) fabricated via in situ polymerisation by Luo et al. [62]. Interestingly, it was found that the intrinsic Young’s modulus of the shell material from FEA simulation (Equation (2)) and the predicted *E*_*FEA**_ (Equation (9)) were 2.92 ± 0.29 GPa and 2.91 ± 0.29 GPa, respectively (validation data shown in Appendix A). The excellent agreement of the two values demonstrates the large applicability of the model to synthetic microparticles with a spherical morphology and a core–shell configuration. Moreover, the average number based diameter and *h/r* of thirty MGF microcapsules were 10.0 ± 0.8 µm and 0.067 ± 0.007, respectively, suggesting a good narrow dispersity and homogeneity of the sample. Similar results were also documented by Mercadé-Prieto et al. [5] for simple MF microcapsules.

In addition, Equation (6) was further validated against composite double coated microcapsules, reported in our previous studies [55]. For fCh-GA microcapsules with an additional maltodextrin based coating, it was determined that the intrinsic Young’s modulus of the overall shell material obtained from FEA simulation was 2.59 ± 0.83 GPa, whereas the *E*_*FEA**_ predicted from the apparent Young’s modulus (Hertz) was 2.61 ± 0.84 GPa (validation data shown in Appendix A). As anticipated, the two values independently generated by FEA and *φ* were in total agreement, demonstrating the effectiveness of the model on double coated microcapsules, which had been fabricated using a two-stage chemical-physical approach (complex coacervation followed by spray drying) [55]. Based on the above, these findings confirm the applicability of the novel model to different types of core–shell microcapsules, independently of the microencapsulation process and shell chemistry. Notwithstanding, when compared to HS laden microcapsules made with a single fCh-GA shell, the mean Young’s modulus value from maltodextrin coated microcapsules is significantly greater. This result indicates the extra maltodextrin coating provided additional stiffness to the shell effectively, thereby enhancing the overall mechanical properties of coacervate shell microcapsules [55]. Overall, these findings not only have demonstrated the superb prediction capability of the newly developed model but have also elucidated its broad spectrum applicability to synthetic, non-synthetic, and composite microcapsules.

## 4. Conclusions

Microcapsules with a liquid perfume core and a fungal chitosan–gum Arabic shell produced via CC were assessed for their intrinsic mechanical properties insightfully. The compression force dataset obtained from micromanipulation measurements based on compression of single microparticles to different deformations was utilised to determine the apparent Young’s modulus of the whole microcapsule (including the liquid core) and the intrinsic Young’s modulus of shell material by Hertz and FEA models, respectively. It was found that the apparent Young’s modulus of whole microcapsules was 0.095 ± 0.014 GPa, whilst the intrinsic *E_FEA_* of the shell of microcapsules was 1.02 ± 0.13 GPa (yielding (*h/r*)*_FEA_* = 0.132 ± 0.009) on average, which differ by about one order of magnitude. The mathematical interrelationship between *E_A_* and *E_FEA_* was determined by numerical simulations resulting in *E_FEA_*/*E_A_* as a third degree of polynomial function of *h/r*. The prediction capability of the newly developed model was validated against a broad spectrum of microcapsules (i.e., synthetic, non-synthetic, double coated) thereby elucidating its general validity regardless of the nature (chemistry) of microcapsules. The new model therefore represents a powerful and rapid tool to determine the intrinsic Young’s modulus of the microcapsule shell material when only the apparent Young’s modulus of whole microcapsules is known, or no FEA simulating tool is available. Moreover, micromanipulation provides a rapid pathway to investigate the mechanical rupture of microcapsules with accuracy and dexterity, which is crucial to ensuring their functionalities at end-use applications. Future work can be directed at developing rigorous software to help predict both structural and mechanical property parameters of microparticles, which may pave an avenue to facilitate the development, engineering, and functionalisation of microcapsules for a wide range of applications.

## Figures and Tables

**Figure 1 micromachines-14-00123-f001:**
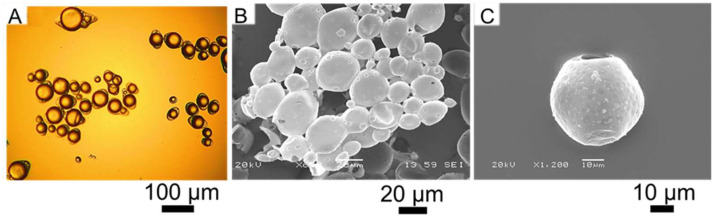
Suspended HS-microcapsules in water by bright-field microscopy (**A**); SEM micrographs of (**B**) HS-microcapsules (overview), (**C**) an incomplete HS microcapsule with surface vacuoles.

**Figure 2 micromachines-14-00123-f002:**
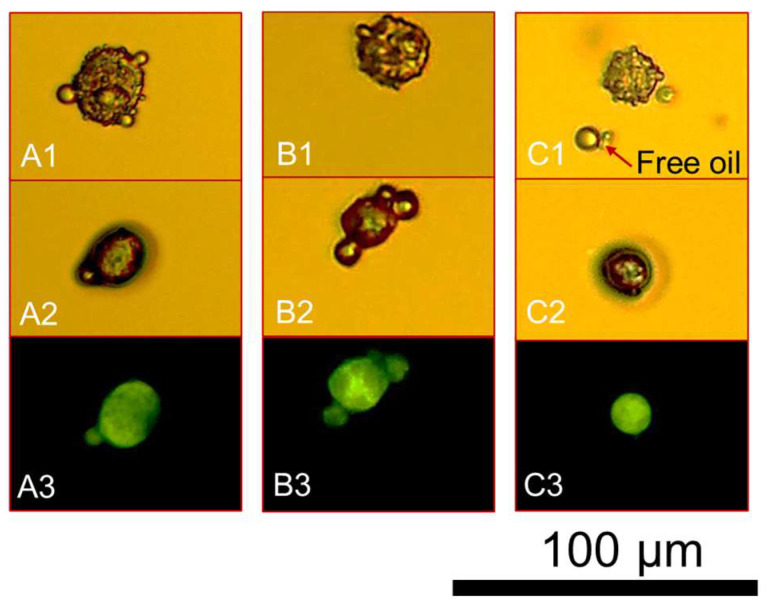
Bright-field optical microphotographs of microcapsules with a spherical morphology suspended in water (**A1**–**C1**); after drying (**A2**–**C2**) and under fluorescence (**A3**–**C3**).

**Figure 3 micromachines-14-00123-f003:**
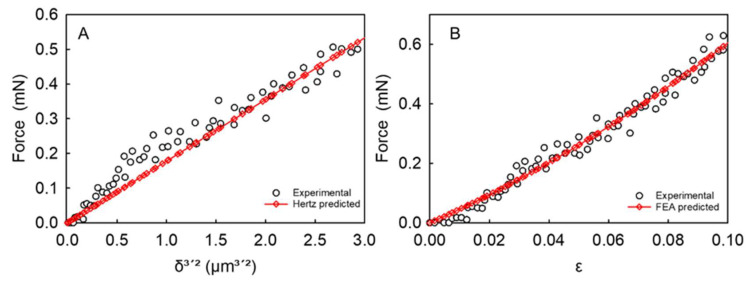
Typical fitting of a (**A**) force-displacement curve by Hertz model (R^2^ = 0.95) and (**B**) force-fractional deformation (*ε* ≤ 0.1) curve by FEA simulation results (R^2^ = 0.98) for a single microcapsule (*d* = 23.7 µm).

**Figure 4 micromachines-14-00123-f004:**
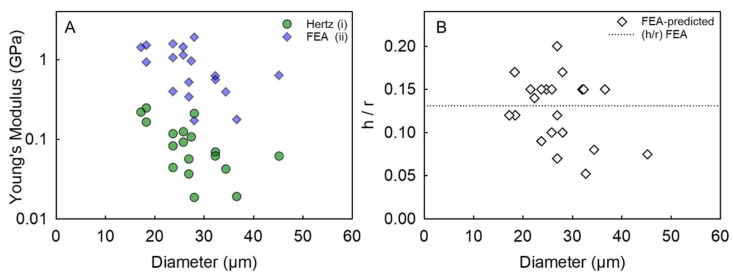
(**A**) Hertzian and FEA-derived Young’s modulus and (**B**) FEA-predicted *h/r* of single microcapsules versus diameter; the dashed line represents the average FEA-predicted (*h/r*)*_FEA_* = 0.132 ± 0.009.

**Figure 5 micromachines-14-00123-f005:**
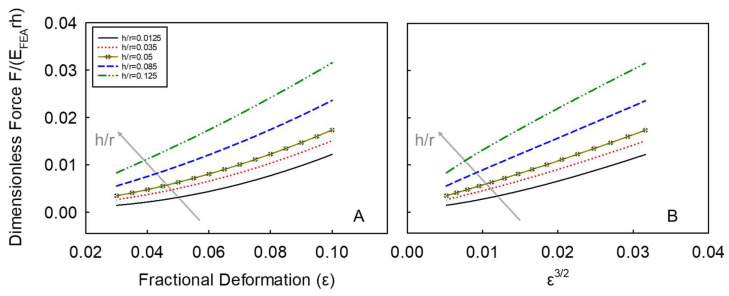
(**A**) Dimensionless force versus fractional deformation data (up to 10% deformation) for a given *h/r*; (**B**) dimensional force versus fractional deformation to a power of 3/2. The simulated data are based on 5 example values of *h/r*, namely 0.0125, 0.035, 0.05, 0.085, 0.125.

**Figure 6 micromachines-14-00123-f006:**
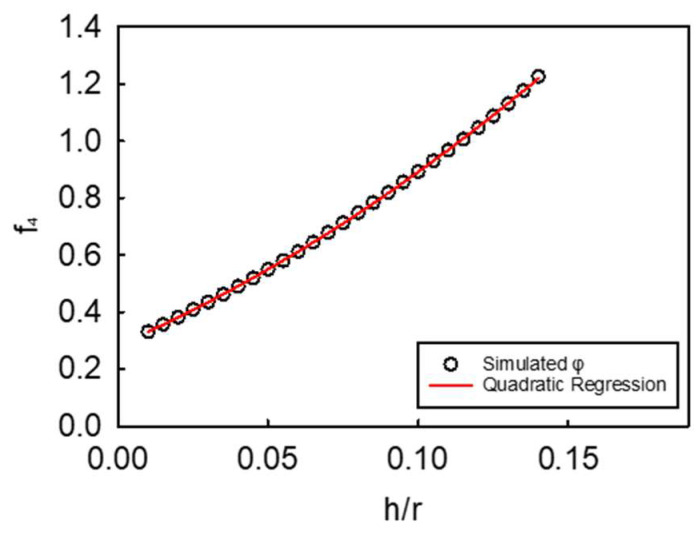
Simulated *f*_4_ function (directly proportional to the scope function *φ*) generated using a series *h/r* values ((*h/r*)_k_ = 0.01, 0.02, …, 0.14) within the applicable domain of the model (*h/r* ≤ 0.14). The circles (o) represent the simulated points as a function of *h/r*, whereas the red line (**–**) is the corresponding second-degree polynomial regression.

**Table 1 micromachines-14-00123-t001:** Complex polynomial functions of *h/r* by Mercadé-Prieto et al. [5].

Coefficient	Polynomial Function
*f* _1_	95071.891 (*h/r*)^5^ − 28426.030 (*h/r*)^4^ + 2411.056 (*h/r*)^3^ − 7.476 (*h/r*)^2^ − 10.829 (*h/r*) + 1.52882
*f* _2_	−318.702 (*h/r*)^4^ + 120.784 (*h/r*)^3^ − 11.380 (*h/r*)^2^ + 2.518 (*h/r*) − 0.05792
*f* _3_	−0.004242 (*h/r*) + 0.00107

**Table 2 micromachines-14-00123-t002:** Intrinsic mechanical property parameters of HS laden microcapsules (Mean ± St. Error). The symbol * represents the predicted intrinsic Young’s modulus via modelling.

Title 1	HS Laden Microcapsules	
Number based diameter (µm)	27.6 ± 1.7	
*E_FEA_* (GPa)	1.02 ± 0.13	R^2^ = 0.98
*E_A_* (GPa)	0.095 ± 0.014	R^2^ = 0.95
*E_A_/E_FEA_ (*GPa*)*	0.085 ± 2 × 10^−3^	
*E_FEA*_* (GPa)	1.09 ± 0.14	
*(h/r)_FEA_*	0.132 ± 9 × 10^−3^	

## Data Availability

Not applicable.

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
