# Peer review of "Relationship between the Young’s Moduli of Whole Microcapsules and Their Shell Material Established by Micromanipulation Measurements Based on Diametric Compression between Two Parallel Surfaces and Numerical Modelling"

_micromachines, 2023, doi:10.3390/mi14010123_

Round 1
Reviewer 1 Report
In the article, the Young’s modulus values of microplastic-free plant based microcapsules with a core of perfume oil (hexyl salicylate) were calculated. Moreover,It is found that the ratio of the two Young’s module values (EA/EFEA) is only a function on the ratio of the shell thickness to radius (h/r) of the individual microcapsule, which can be fitted by a third-degree polynomial function of h/r. This study is interesting and the experimental analysis is also detailed. For the author's work, I have the following suggestions:
(1) It is best to summarize the contributions of this article in the Introduction section.
(2) In the title, the authors highlight the use of micromanipulation to study the Young's modulus of microparticles, but after reading the article, I feel that micromanipulation is simply a tool to generate micro forces, and the content of the article is not consistent with the title. The micromanipulation tools used by the authors can indeed generate microforces, but from a quantitative point of view, how does the micromanipulative tool affect the measurement of Young's modulus? it is best to explain the relationship between the two.
Reviewer 2 Report
1. The authors should read carefully the 'Guide for authors' of the journal to refine this manuscript, as there are too many non-standard formatting issues. For example, the fonts in formula (6) and in the context are different.
2. The literature review is inadequate.
3. Some comparison results with the conventional methods should be given.
4. The author should elaborate detailedly the innovations and research gaps.
5. The conclusion part needs to be improved.
6. The word 'where' after formula (1) does not need to be indent.
7. The word 'which' after formula (7) does not need to be indent.
8. h/r should be italic.
Round 2
Reviewer 2 Report
All the problems have been solved.